# Lift-Off Ablation of Metal Thin Films for Micropatterning Using Ultrashort Laser Pulses

**Byunggi Kim, Han-Ku Nam** **, Young-Jin Kim and Seung-Woo Kim ***

Department of Mechanical Engineering, Korea Advanced Institute of Science and Technology,
Daejeon 34141, Korea; bkim2017@kaist.ac.kr (B.K.); hanku.nam@kaist.ac.kr (H.-K.N.); yj.kim@kaist.ac.kr (Y.-J.K.)
* Correspondence: swk@kaist.ac.kr

**Abstract:** Laser ablation of metal thin films draws attention as a fast means of clean micropatterning. In this study, we attempt to remove only the metal thin film layer selectively without leaving thermal damage on the underneath substrate. Specifically, our single-pulse ablation experiment followed by two-temperature analysis explains that selective ablation can be achieved for gold (Au) films of 50–100 nm thickness by the lift-off process induced as a result of vaporization of the titanium (Ti) interlayer with a strong electron–phonon coupling. With increasing the film thickness comparable to the mean free path of electrons (100 nm), the pulse duration has to be taken shorter than 10 ps, as high-temperature electrons generated by the ultrashort pulses transfer heat to the Ti interlayer. We verify the lift-off ablation by implementing millimeters-scale micropatterning of optoelectronic devices without degradation of optical properties.

**Keywords:** thin film ablation; femtosecond pulse laser; interfacial heat transfer; electron–phonon coupling



## 1. Introduction

Ultrashort pulse lasers are being widely investigated as an advanced tool for laser ablation of material with reduced thermal damage and improved patterning resolutions [1–3]. The laser pulse duration in the picosecond regime permits light energy absorption and subsequent removal by vaporization from material to take place ahead of the onset of lattice thermal conduction, thereby reducing excessive energy accumulation outside the area of laser incidence. Such ultrafast laser ablation is exploited to minimize the heat-affected zone on diverse materials including metals, semiconductors, and biomaterials [2]. Furthermore, ultrashort lasers can readily provide extremely high peak intensities, triggering extraordinary nonlinear behaviors of electron dynamics within material [4], leading to various nonconventional schemes of materials processing. Sub-diffraction-limited patterning with a 40 nm linewidth was demonstrated by excitation of ultraviolet femtosecond laser-induced surface plasmon polaritons [5]. In addition, temporally-shaped ultrashort pulses were applied for debris-less drilling with a diameter of 300 nm [6,7], while control of electron dynamics was extended to high aspect ratio deep drilling in silicon [8,9] and dielectrics [10,11].

Nowadays there is much interest in micropatterning of metal thin films using ultrashort lasers, particularly for industrial applications such as solar cells scribing [12,13] and display devices repairing [14] with submicron variation in linewidth. Such microelectronic devices consist of metallic film layers deposited on a semiconductor or dielectric substrate, so heat transfer through the interface between different material layers plays an important role in temperature evolution inside thin film layers [15,16]. Besides, ultrashort intense laser pulses cause a nonequilibrium state within electrons and phonons, thereby introducing exceptional routes of heat transfer in the form of electron–electron scattering and electron–phonon coupling. Consequently, understanding ultrafast thermodynamics across thin film layers becomes essential to implement clean and selective ablation of thin film devices.

In this study, we investigate the lift-off process of metal thin films, first proposed as a fast means of clean micropatterning to replace noxious wet or dry etching [17]. In the framework, we perform experimental and numerical simulation to analyze the effect of electron–electron and/or electron–phonon coupling of the metal film layer and the interlayer on the lift-off process, through a comparison between a 100 nm-thick gold (Au) film and a chromium (Cr) film of the same thickness but having a different electron mean free path. We also extend our investigation to find how the laser pulse duration affects the substrate damage. Finally, we apply the lift-off process to actual selective ablation for large-area patterning of optoelectronic devices.

## 2. Method: Single-Pulse Ablation Experiment and Analysis

### 2.1. Principle of Lift-Off Process

Figure 1 illustrates the lift-off process enabled using a single ultrashort pulse of 10 ps or shorter duration. The conductive metal layer provides a strong electron–electron coupling at the metal–interlayer interface so as to increase electron temperature of the interlayer. For 0.2 ps pulse ablation, the lift-off process is realized by ultrafast vaporization of the interlayer with a strong electron–phonon coupling. Electronic heat transfer in a nonequilibrium electron–phonon state triggers the lift-off process, affecting the ablation threshold and surface morphology. On the other hand, for 100 ps pulse ablation, heat steadily dissipates into the substrate via the interlayer without inducing the lift-off process with significant substrate damage.

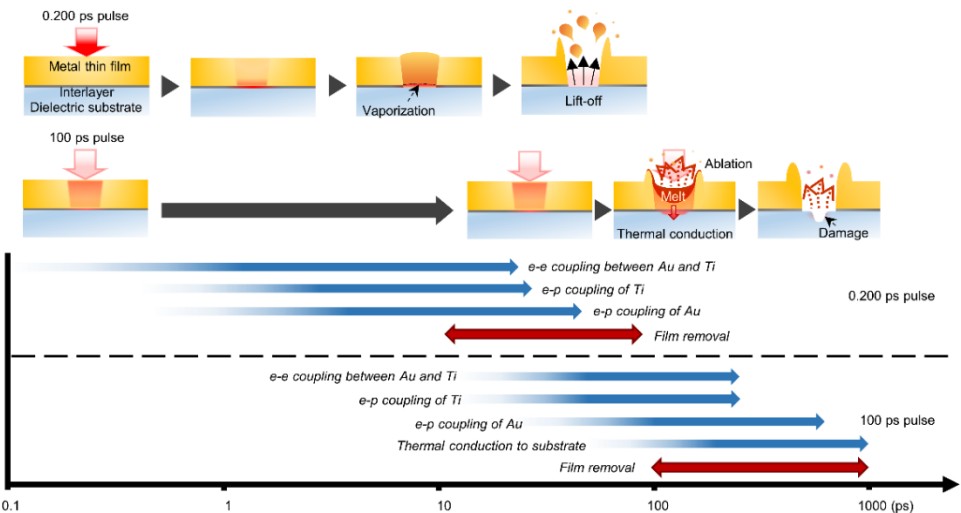

**Figure 1.** Schematic illustration of the ablation process of a thin metal film (Au) in the pulse duration range of 0.2–100 ps, reproduced from [17], with permission from Springer Nature, 2021.

### 2.2. Experimental Setup

Figure 2 illustrates our experimental setup configured for micropatterning of metal thin films on dielectric substrates. An ytterbium-doped fiber femtosecond laser with a 1035 nm center wavelength is amplified by chirped-pulse amplification, while its pulse duration is adjusted in the range of 0.2~84.6 ps. The pulse repetition rate of the source laser is set at 200 kHz. Pulse picking is made by combining an electro-optic modulator (EOM) with a polarizing beam splitter (PBS) under timing control using an arbitrary waveform generator (AWG) [17]. Experimental samples are prepared by depositing Au and Cr thin films on a glass substrate, with a Ti interlayer being inserted as an adhesion layer to Au films. A three-axis motorized stage is used to move the sample, with the *z*-direction being controlled so as to ensure the focal point of the objective lens is always kept on the sample surface.

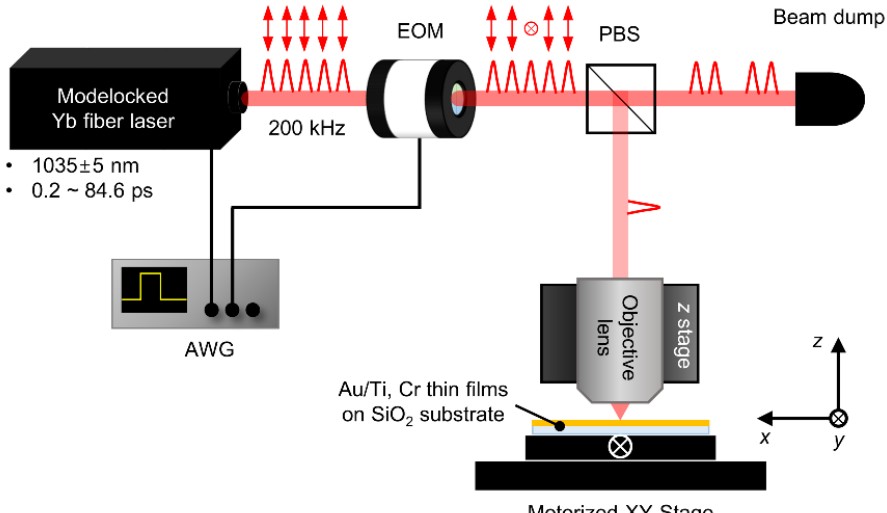

**Figure 2.** Schematic illustration of experiment setup. The laser pulse duration is adjusted in the range of 0.2 to 84.6 ps. Single-pulse picking is achieved through an electro-optic modulator (EOM) in combination with a polarizing beam splitter (PBS) and an arbitrary waveform generator (AWG). The metal thin film sample is positioned using a 3-axis motorized stage.

A scanning electron microscope (SU5000, Hitachi, Daejeon, Korea) was used to image the micropatterned samples. The craters' depth made by a single laser pulse was measured with a confocal microscope (VK-X1000, Keyence, Daejeon, Korea). For each processing condition, the depth value was determined by averaging 10 craters along with its standard deviation.

*2.3. Two-Temperature Model*

Irradiation of ultrashort pulses introduces a thermally nonequilibrium state between electrons and phonons in material. Thus, two-temperature modeling is necessary to analyze the heat transfer between the electronic and lattice subsystems [18–20]. Here, we briefly explain the two-temperature model adopted in this investigation. The electron temperature ($T_e$) and lattice temperature ($T_l$) in the transient regime are described in the form of electron–phonon coupling as:

$$C_e \frac{\partial T_e}{\partial t} = \nabla(k_e \nabla T_e) - G_{e-p}(T_e - T_l) + S \tag{1}$$

$$\rho_l \left[ c_{pl} + L_m \delta_m + L_v \delta_v \right] \frac{\partial T_l}{\partial t} = \nabla(k_l \nabla T_l) + G_{e-p}(T_e - T_l) \tag{2}$$

where $C_e$, $k$, $G_{e-p}$, $\rho_l$, and $c_{pl}$ denote the specific heat of free electron, thermal conductivity, electron–phonon coupling coefficient, lattice density, and specific heat of lattice, respectively. The latent heat $L$ is included in numerical calculation by using the Kronecker delta-like function $\delta$. The heating term $S$ accounts for the electron temperature increase induced by absorbing a laser pulse. Now, assuming a spatiotemporally Gaussian profile, the heating term $S$ is given as

$$S = \frac{1-R}{1-\exp\left[-\frac{L}{(d+d_b)}\right]} \times 4 \times \sqrt{\frac{4\ln 2}{\pi}} \frac{F}{(d+d_b)t_p}$$
$$\times \exp\left[-\frac{z}{d+d_b} - 4\ln 2\left(\frac{t-2t_p}{t_p}\right)^2\right] \tag{3}$$

where $R$, $L$, $F$, and $t_p$ are the reflectivity, film thickness, laser fluence, and pulse duration, respectively. The optical penetration depth $d$ is a reciprocal of the absorption coefficient. Here, the mean free path of electrons $d_b$ can be considered as an extension of the optical

penetration depth, as high-temperature electrons rapidly distribute thermal energy within an extremely short period of ~100 fs [21,22].

We considered two paths of interfacial heat transfer: the electron–electron coupling and the phonon–phonon coupling. The thermal resistance of the electron–electron coupling $R_{ee}$ defined at a metal–metal interface is given as [15,23]

$$R_{ee} = \frac{4(Z_A + Z_B)}{Z_A Z_B} \tag{4}$$

where $Z_A = C_{e,A} v_{e,A}$ and $Z_B = C_{e,B} v_{e,B}$. Subscripts A and B denote different kinds of metals. The electron velocity $v_e$ is assumed being close to the Fermi velocity. The thermal resistance of the phonon–phonon coupling $R_{pp}$ is defined at both the metal–metal and metal–dielectric interfaces. We set $R_{pp}$ to be equal to $4 \times 10^{-9}$ m$^2$ K/W [15,24]. On the surface, all heat transfer paths such as natural convection, radiation, and thermal conduction to air are neglected. The thermophysical properties used in our simulation are summarized in [17].

## 3. Results and Discussion

### 3.1. Effect of Electron Mean Free Path of Metal Film

Figure 3a,b show SEM images of the craters fabricated by single-pulse irradiation on 100 nm-thick Au and Cr thin films with varying the pulse duration. It is noted that the Au film experiences complete removal by lift-off with irradiation of sub-10 ps pulses. In contrast, the Cr film shows no sign of lift-off for the whole range of pulse durations; it suffers nanoscale surface irregularities and microcracks caused by phase explosion at liquid–solid interface followed by rapid cooling [25]. The experimental observations are verified by the 2D lattice temperature profiles of Figure 3c,d, calculated using the two-temperature model described in the previous chapter, inside the Au and Cr films at the moment the temperature reaches its maximum upon irradiation with a 0.2 ps pulse. In comparison, the Au film permits a larger amount of heat to be transferred along the thickness direction with a large ~100 nm electron mean free path. Figure 3e shows the ablation threshold as a function of film thickness. The Au film requires higher fluence over 5.2 J/cm$^2$ to increase the lattice temperature with increasing the film thickness. With the help of a large electron mean free path of the Au metal layer, heat transports to the Ti interlayer via electron–electron coupling, thereby initiating vaporization followed by lift-off ablation. On the other hand, the Cr film has a short mean free path of ~16 nm, so the surface begins to melt before heat arrives at the bottom even at a much smaller laser fluence of 0.52 J/cm$^2$. This simulation result indicates that ultrafast electron–electron coupling with a large mean free path plays an important role in the lift-off ablation of metal thin films. Nonetheless, for longer pulse durations over 10 ps, the temperature of the Ti interlayer cannot reach the vaporization point even with large laser fluence because of too much heat dissipation within the metal film layer.

Figure 3f shows the crater diameter as a function of pulse duration. The crater dimeter was calculated under the assumption that ablation occurs when the whole metallic film layer melts or the Ti interlayer vaporizes. The crater depth has a minimum value around the 10 ps pulse duration, where the ablation mechanism changes from the lift-off process to the thermal ablation. Heat is dissipated to the radial direction via scattering of high-temperature electrons for shorter pulse durations, while thermal conduction becomes significant with increasing the pulse duration. As a result, a picosecond pulse delivers a better spatial resolution compared with a femtosecond pulse. Our results confirm that the temporal pulse profile has significant effects on the machining resolution of metal thin films.

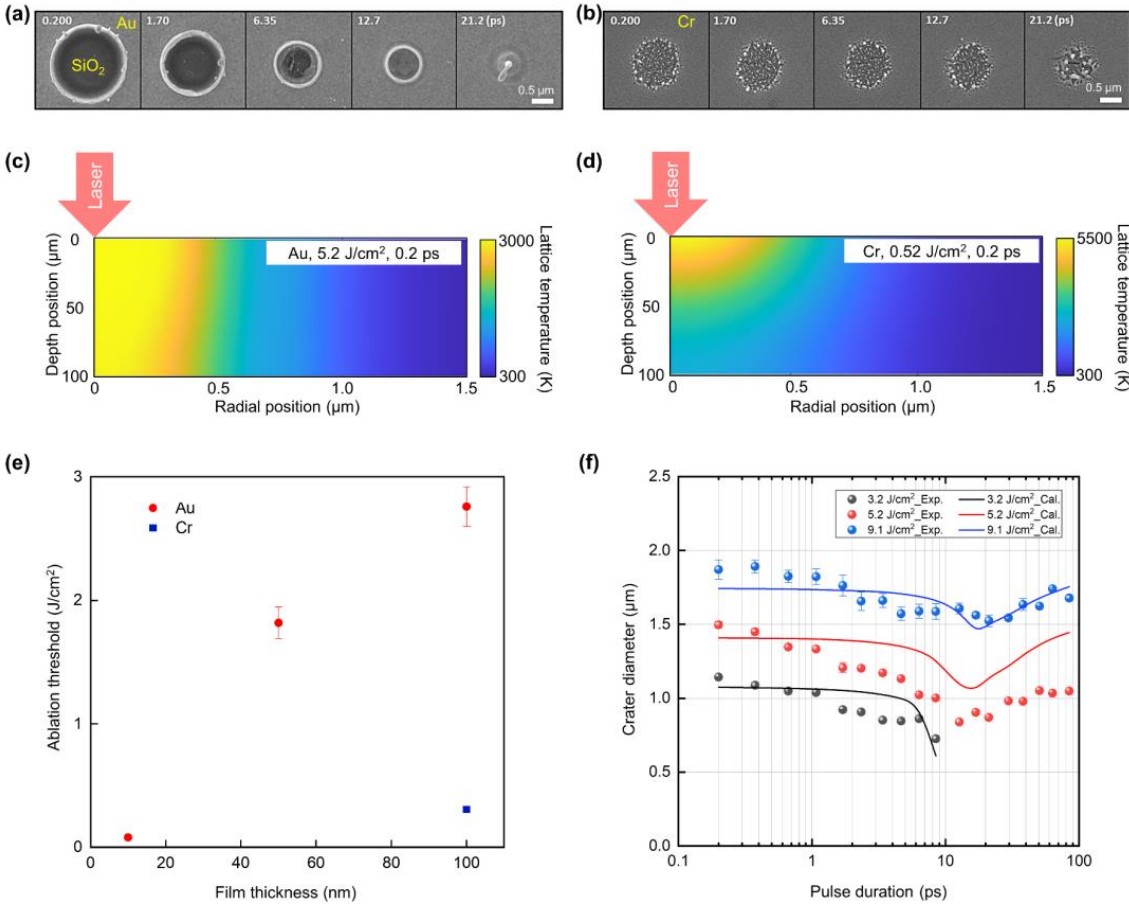

**Figure 3.** Single-pulse ablation on 100 nm-thick Au and Cr films. SEM images of the craters made on (**a**) Au and (**b**) Cr thin films with laser fluence of 5.2 and 0.52 J/cm², respectively. Pulse durations are indicated in the SEM images. Calculated 2D temperature distribution inside (**c**) Au and (**d**) Cr thin films. (**e**) Ablation threshold of Au and Cr films as a function of thickness. (**f**) Crater diameter as a function of pulse duration.

### 3.2. Selective Ablation Induced by Lift-Off Process

Figure 4a,b present experimental results on the depth distribution of the craters on 50 nm- and 100 nm-thick Au films, respectively, obtained with varying the laser fluence and the pulse duration. As measured with a confocal microscope, the crater depth on the 50 nm-thick Au film yields a steadily rising tendency with increasing the laser fluence, but not strongly affected by the pulse duration. Notably, for a particular fluence of 2.6 J/cm², the Au film experiences a selective removal within a 5 nm deviation for all the pulse durations employed in the experiment. On the other hand, for the 100 nm-thick Au film, selective lift-off is achieved only when the pulse duration is shorter than 10 ps, for a fluence of 3.2 J/cm². For larger fluences, substantial amount of damage on the glass substrate is observed especially for pulse durations longer than 10 ps.

Figure 5 illustrates the temperature distribution calculated along the thickness direction at the moment of ablation using the two-temperature model with varying the pulse duration in the range of 0.2–84.6 ps. For the 50 nm-thick Au film, the substrate damage due to the temperature rise above the glass transition threshold is found to be not significant, less than 15 nm for all the pulse durations. However, in the case of 100 nm-thick film, the substrate damage develops for longer pulse durations because the Ti interlayer cannot reach the vaporization point due to too much heat dissipation in the metal film. This implies that the pulse duration needs to be shortened to accelerate the Au/Ti interfacial heat transfer based on electron–electron coupling for selective lift-off ablation.

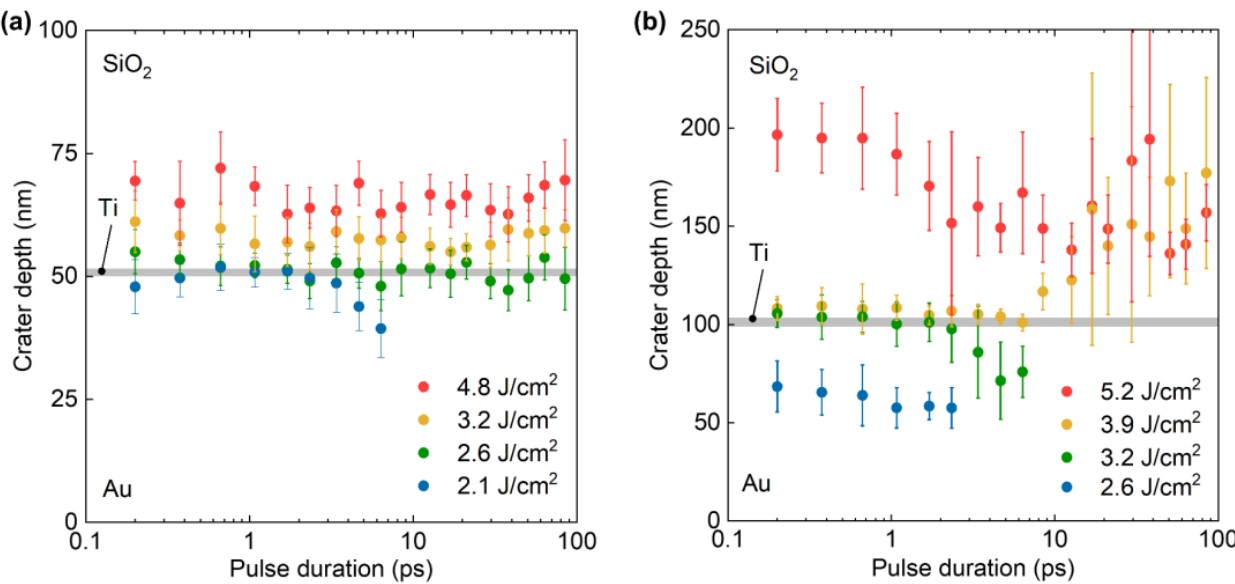

**Figure 4.** Distributions of crater depths fabricated by single-pulse ablation as a function of the pulse duration. (**a**) 50 nm-and (**b**) 100 nm-thick Au films; (**b**) has been reproduced from [17] under permission of Springer Nature, 2021. Standard deviations for 10 craters are indicated as error bars. The range of pulse duration shorter than 10 ps is found suitable for selective ablation of 100 nm-thick Au film.

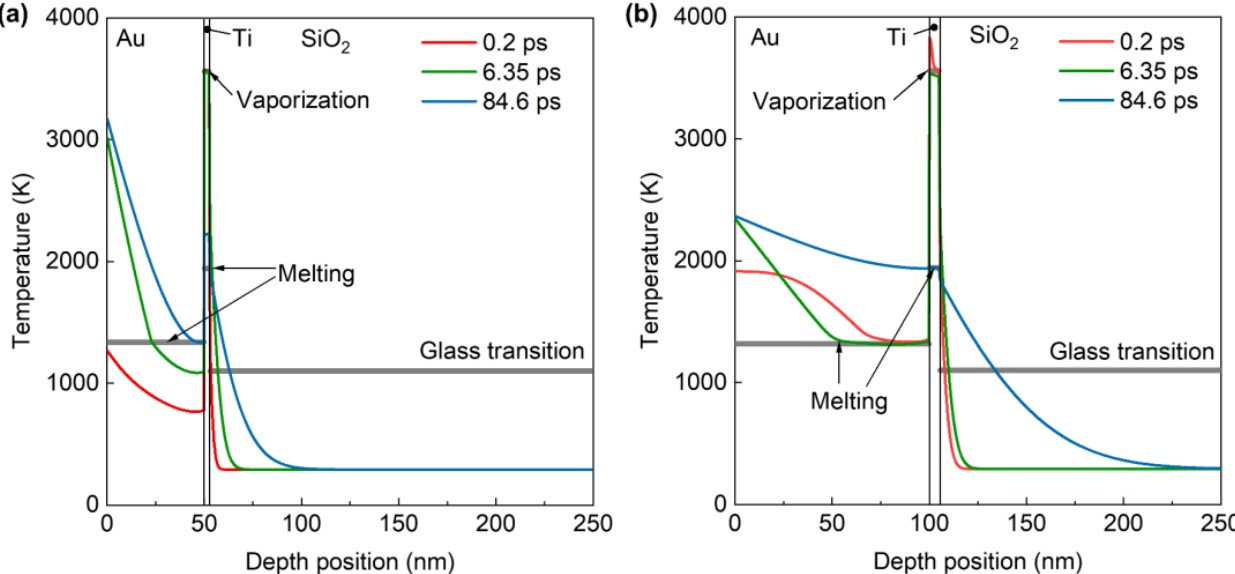

**Figure 5.** Lattice temperature distribution along the depth position at the moment of ablation. (**a**) 50 nm- and (**b**) 100 nm-thick Au films. For 100 nm-thick Au film, significant amount of heat is conducted to the glass substrate for long pulse duration to induce glass transition.

We consider the ballistic motion of electrons that is known to strongly affect the electron–electron coupling at metal–metal interface, as an additional light penetration depth $d_b$ in Equation (3). This permits our dynamic analysis of temperature development to predict reasonably well the physical mechanism of our selective lift-off ablation. Nonetheless, given the fact that electron and thermal dynamics induced by ultrashort pulses are highly complicated phenomena, further improvements are necessary so that theoretical analysis can better describe the interfacial heat transfer. Specifically, the thermodynamic properties and interfacial thermal resistance in the nonequilibrium state have to be more accurately estimated by incorporating the semiclassical Boltzmann transport

theory [26–28]. This will bring more controllability on electron dynamics by means of ultrashort optical pulses for advanced ablation schemes.

### 3.3. Applications in Optoelectronic Devices Fabrication

In our previous study [17], Raman spectrum analysis revealed that the glass substrate undergoes a minimal modification. In addition, the optical transmittance of the glass substrate was conserved after large area patterning using 0.2 ps pulses, while critical deterioration was observed for 4 ns-pulse ablation. In this study, we applied the lift-off ablation to the microfabrication of optoelectronic devices, as shown in Figure 6. First, two transmission diffraction gratings were fabricated on a 100 nm-thick Au thin film on a glass substrate; with insertion of a 5 nm-thick Ti interlayer, one was created in a line shape (Figure 6b) and the other in a dots array (Figure 6c). The diffraction efficiency of the two gratings were measured by conducting diffraction imaging using a helium–neon laser (Figure 6a). In addition to the diffraction images (Figure 6d,e), quantitative analysis indicates that the gratings made with a 0.2 ps laser pulse yields a strong first-order diffraction, reaching 60% of the zeroth-order diffraction (Figure 6f). For comparison, when the same gratings were made using a conventional 4 ns pulse laser, its first-order diffraction was just ~15% of the zeroth-order diffraction, evidencing a relatively poor accuracy of grating patterning. Lastly, a honeycomb-structured metal grid was patterned to fabricate a flexible transparent–conductive layer (Figure 6g). A 100 nm thick Au film was deposited on a 1 mm thick polyethylene terephthalate (PET) as well as a 50 μm thick sapphire ($Al_2O_3$) substrate with a 25 nm thick indium tin oxide interlayer and a 5 nm thick Ti interlayer, respectively. The flexible transparent–conductive devices showed a minimal degradation of light transmittance and mechanical strength. Figure 6h,i present expanded views of Au grids formed on the PET and $Al_2O_3$ substrates, respectively. The results confirmed that damage-free patterning can be achieved for 100 nm thick Au films on thermally vulnerable substrate materials such as polymer (melting temperature of PET: 533 K) and thin dielectrics as the lift-off process suppresses heat transfer to substrate materials.

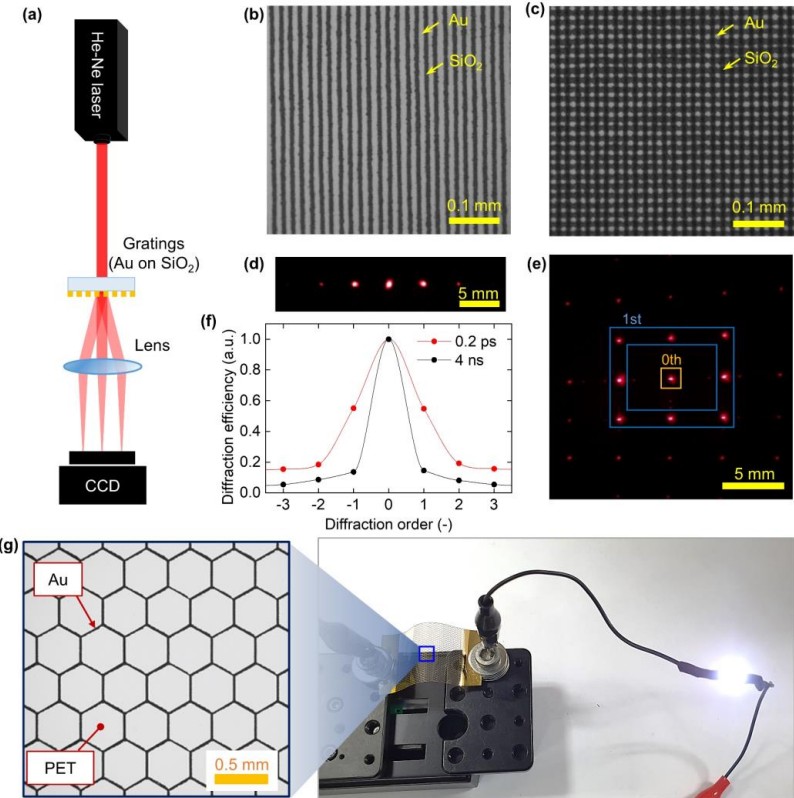

**Figure 6.** *Cont.*

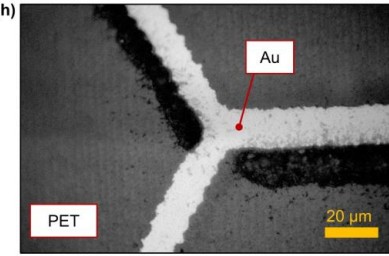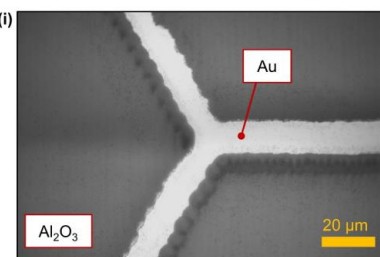

**Figure 6.** Optoelectronic devices fabrication using ultrashort laser pulses. (**a**) Diffraction imaging of transmission gratings by using a He–Ne laser. The sample gratings were fabricated with 0.2 ps laser pulses on a 100 nm thick Au film deposited on glass substrate. Optical microscope images of line-shaped (**b**) and dots array (**c**) transmission gratings. Diffraction images from the line-shaped grating (**d**) and dots array grating (**e**). (**f**) Diffraction efficiency of the line-shaped grating. Comparison is made with the same grating fabricated with a 4 ns laser. (**g**) Transparent conductive honeycomb structure patterning on a 100 nm thick Au on a flexible PET substrate. (**h,i**) Expanded optical microscope images of metal grids fabricated on PET and $Al_2O_3$ substrate, respectively. Scale bars in (**d,e**), (**g**), and (**h,i**) indicate 5 mm, 0.5 mm, and 20 μm, respectively.

## 4. Conclusions

Thermophysical mechanism of the ultrafast laser ablation enabling selective lift-off removal of metal thin films from the underneath dielectric substrate was investigated for the purpose of establishing a fast tool of micropatterning using ultrashort pulse lasers. This investigation revealed that the intended selective lift-off processing can be achieved by inserting a Ti interlayer that triggers explosive vaporization by ultrafast heating with a strong electron–phonon coupling in addition to an efficient electron–electron coupling with the metal film layer. This ultrafast ablation of using ultrashort laser pulses may be further refined for more efficient processing of metal thin films; for example, the temporal profile of ultrashort pulse train can be manipulated to improve heat transfer into the interlayer with consideration of the transient change of quantum coupling among electrons and phonons. Furthermore, the ultrafast interfacial heat transfer may be reflected in the design phase of thin films structures in view of lift-off ablation processing. Finally, it is expected that our study will lay a foundation for further development of advanced processing of metal thin film devices, particularly for optoelectronic industrial applications.

**Author Contributions:** Conceptualization, B.K. and S.-W.K.; methodology, B.K.; software, B.K.; validation, B.K., H.-K.N., Y.-J.K. and S.-W.K.; formal analysis, B.K. and H.-K.N.; investigation, B.K. and H.-K.N.; data curation, B.K. and H.-K.N.; writing—original draft preparation, B.K.; writing—review and editing, S.-W.K.; project administration, Y.-J.K. and S.-W.K.; funding acquisition, Y.-J.K. and S.-W.K. All authors have read and agreed to the published version of the manuscript.

**Funding:** This research was funded by National Research Foundation of the Republic of Korea (NRF-2012R1A3A1050386, NRF-2020R1A2C2102338, NRF-2021R1A4A1031660).

**Institutional Review Board Statement:** Not applicable.

**Informed Consent Statement:** Not applicable.

**Data Availability Statement:** The data presented in this study are available on request from the corresponding author.

**Acknowledgments:** Authors appreciate KAIST Analysis Center for Research Advancement (KARA) and Korea Institute of Machinery and Materials (KIMM) for supports in analysis of experimental data.

**Conflicts of Interest:** The authors declare no conflict of interest. The funders had no role in the design of the study; in the collection, analyses, or interpretation of data; in the writing of the manuscript, or in the decision to publish the results.

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
