# Peer review of "Lift-Off Ablation of Metal Thin Films for Micropatterning Using Ultrashort Laser Pulses"

_metals, doi:10.3390/met11101586_

Round 1

Reviewer 1 Report

This work investigated the selective lift-off ablation of metal thin films from the underneath dielectric substrate and its thermophysical mechanism of the ultrafast laser ablation. This work has presented some interesting for publication. However, further improvements are needed to enhance the quality of the manuscript for reconsideration. The comments are shown as follows.

  1. Abstract contains too many backgrounds but less results. Please do the other way.
  2. Introduction: please add more research progress (with results, rather generic descriptions) in the relevant work.
  3. Figure 1 and related descriptions should be in Section 2.
  4. The characterization methods and facilities are missing. These should be described in Section 2.
  5. Figure 3a and b: there is no scale bars.
  6. although it is interesting to see the Applications in optoelectronic devices fabrication (Figure 6), there is no detailed results on the microstructure and properties of the produced. It would be more interesting if the authors could provide such results. If this cannot be provided, it is highly recommended to alter some more parameters (e.g. thickness of the Au film, other processing parameters) to investigate their effects on processing.

Reviewer 2 Report

In general, this is a nice study for the development of selective thin-film removal in microfabrication processes.

One problem that I found in this manuscript is that many of the information in the present manuscript has been published by the authors in their previous work. See Ref. 17.   

In the present manuscript, the author only added 50 nm-thick Au film and compared it with the result of 100 nm-thick Au from their previous work to show the dependence of the ablation threshold fluence on the thickness of the Au film (and pulse durations). I felt that this is not enough to justify another publication. Numerous previous studies have shown that ablation is strongly dependent on the laser fluence, wavelength, pulse duration, film thickness, etc. What is not clear is how the dependence will look like. So, in my opinion, the present manuscript in the current form does not offer new knowledge or insights.  

In the previous work, the author reported that significant substrate damage can be avoided with 3.9 J/cm2 laser fluence and pulse duration less than 10 ps. So, I would suggest beefing up the present manuscript with additional data. For example, the dependence of crater depth as a function of film thickness (at least for different thicknesses) at the optimum laser fluence with pulse duration of 10 ps, or the dependence of threshold fluence as a function of film thickness with pulse duration of 10 ps.

Many of the figures in the present manuscript are stripped-down version of the figures in their previous work. I understand that the authors. This stripping down process made the images difficult to understand. I would suggest using the published images and ref them instead.

It would be good to also include high-resolution SEM images of the craters.

Again, this is a great preliminary study, but I felt that publication in the present form would be a bit too premature. 

Round 2

Reviewer 1 Report

The revision is satisfactory for publication.

Reviewer 2 Report

Thank you for addressing my comments. I feel the revised manuscript has been significantly improved. I have no further concerns. The revised manuscript is suitable for publication in the journal.